# INFLUENCE-BASED REINFORCEMENT LEARNING FOR INTRINSICALLY-MOTIVATED AGENTS

## ABSTRACT

Discovering successful coordinated behaviors is a central challenge in Multi-Agent Reinforcement Learning (MARL) since it requires exploring a joint action space that grows exponentially with the number of agents. In this paper, we propose a mechanism for achieving sufficient exploration and coordination in a team of agents. Specifically, agents are rewarded for contributing to a more diversified team behavior by employing proper intrinsic motivation functions. To learn meaningful coordination protocols, we structure agents' interactions by introducing a novel framework, where at each timestep, an agent simulates counterfactual rollouts of its policy and, through a sequence of computations, assesses the gap between other agents' current behaviors and their targets. Actions that minimize the gap are considered highly influential and are rewarded. We evaluate our approach on a set of challenging tasks with sparse rewards and partial observability that require learning complex cooperative strategies under a proper exploration scheme, such as the StarCraft Multi-Agent Challenge. Our methods show significantly improved performances over different baselines across all tasks.

## 1 INTRODUCTION

Deep Reinforcement Learning (DRL) has been applied to solve various challenging problems, where an agent typically learns to maximize the expected sum of extrinsic rewards gathered as a result of its actions performed in the environment (Sutton et al., 1998). Multi-Agent Reinforcement Learning (MARL) refers to the task of training a set of agents to maximize collective and/or individual rewards, while existing in the same environment and interacting with each other.

Recent works have shown that agents with coordinated behaviors learn remarkably faster (Roy et al., 2019) since coordination helps the discovery of effective policies in cooperative tasks. Nevertheless, achieving coordination among agents still remains a central challenge in MARL (Jaques et al., 2019). Prominent works often resort to a popular learning paradigm called Centralized Training with Decentralized Execution (CTDE) (Lowe et al., 2017; Foerster et al., 2018), where each agent is evaluated using a centralized critic and has access to extra information about the policies of other learning agents during training. At the time of execution, policies' actions are restricted to local information only (i.e. their own observations). Our work is primarily motivated by the following natural questions:

*In multi-agent settings, how can we quantify the effect of an agent (or player) on other teammates' behaviors (particularly on their performance)? And to what degree exploiting this quantity can lead to coordinated behaviors and consequently better overall performances?*

To that end, we propose a novel approach that aims at promoting coordination for cooperative tasks by augmenting CTDE MARL main return-maximization objective with an additional multi-agent objective that acts as a policy regularizer; we refer to the latter objective as the $influence\ function$. To build intuition, a chosen agent, which we call the "influencer", assesses the progress that other agents are making given its current policy and consequently learns behaviors that will result in an improved performance of its teammates. Concretely, we formulate the influence of an influencer $\pi$ as an estimation of the dissimilarity between other' behaviors and their targets given the current behavior of $\pi$. The influencer is encouraged to learn behaviors that are expected to minimize that

dissimilarity. We also propose two approaches to estimate the influence and empirically show that they yield unbiased estimates of the true value.

Agents acting upon the proposed coordination paradigm learn to efficiently exploit the observed joint action space using available information. However, and since the joint space grows exponentially with the number of agents, it is highly unlikely that agents will have access to sufficient information to learn optimal behaviors to solve the task at hand; this problem arises in many scenarios such as sparse-reward environments, thus a proper exploration scheme is often required. However, many existing multi-agent deep reinforcement learning algorithms still use mostly noise-based techniques (Liu et al., 2021; Rashid et al., 2018; Yang et al., 2018). Moreover, independent exploration proved to be inefficient in cooperative settings (Roy et al., 2019). Recently, this challenge was addressed through Intrinsic Motivation (IM) (Jaques et al., 2019; Du et al., 2019; Zhou et al., 2020b). Many approaches employ IM to encourage exploration of state-space (Han et al., 2020; Burda et al., 2019) or state-action space (Fayad & Ibrahim, 2021) by identifying novel configurations and rewarding an agent for visiting them. We provide an extension of these ideas into multi-agent settings and further build connection between reward shaping and coordinated behavior learning, where we choose an agent to act as an influencer (i.e. regularize its standard objective using the influence function) while other agents learn to maximize the expected sum of both extrinsic and intrinsic rewards.

To sum up, our main contributions are threefold: 1) developing an influence function to promote learning coordinated behaviors and improve team performance; 2) extending exploration via random network distillation to multi-agent settings by crafting a "novelty" function that rewards under-explored behaviors; 3) formulating a novel intrinsic incentive to promote learning diverse team behaviors to help uncover complex behaviors in a collaborative way.

We demonstrate the effectiveness of our methods on a comprehensive set of challenging tasks which include, but not limited to, the StarCraft Multi-Agent Challenges (SMAC) (Samvelyan et al., 2019) and the Multi-Agent Particle Environments (MAPE) (Mordatch & Abbeel, 2018; Lowe et al., 2017). Empirical results show a significant improvement over a wide variety of state-of-the-art MARL approaches. We also conduct insightful ablation studies to understand the relative importance of each component of the approach individually.

## 2 BACKGROUND

### 2.1 MARKOV GAMES

Markov games (Littman, 1994; Agarwal et al., 2020; Daskalakis et al., 2021) are a superset of Markov decision process (MDPs) and matrix games, including both multiple agents and multiple states. Formally, we consider a setting with $N > 2$ agents ($[N] := \{1, 2 \ldots, N\}$) who repeatedly select actions in a shared MDP. The goal of each agent is to maximize their respective action value function. Formally, an MDP is defined as a tuple $G = (S, N, \{A_i, r_i\}_{i \in [N]}, T, \gamma, \rho)$, where:

- $S$ is the finite states space, where the initial states are determined by a distribution $\rho : S \to [0, 1]$. Since we consider partially observable testing environments, we denote by $s_i$ the information observed by agent $i$ about the global state $s \in S$.
- $A_i$ is a finite action space for agent $i$ with generic element $a_i \in A_i$. Using common conventions, we will write $A = \Pi_{j \in [N]} A_j$ and $A_{-i} = \Pi_{j \neq i} A_j$ to denote the joint action spaces of all agents and of all agents other than $i$ with generic elements $\mathbf{a} = (a_i)_{i \in [N]}$ and $\mathbf{a}_{-i} = (a_j)_{i \neq j \in [N]}$, respectively. According to this notation, we have that $\mathbf{a} = (a_i, \mathbf{a}_{-i}) \forall i$.
- $r_i : S \times A \to \mathbb{R}$ is the individual reward function of agent $i$, i.e., $r_i(s, a_i, \mathbf{a}_{-i})$ is the instantaneous reward of agent $i$ when agent $i$ takes action $a_i$ and all other agents take actions $\mathbf{a}_{-i}$ at state $s \in S$.
- $T$ is the transition probability function, for which $T(s'|s, \mathbf{a})$ is the probability of transitioning from $s$ to $s'$ when $\mathbf{a} \in A$ is the action profile chosen by the agents.
- $\gamma$ is a discount factor for future rewards of the MDP, shared by all agents.

In a Markov game, each agent is independently choosing actions and receiving rewards. Conventionally, an agent $k$ aims to maximize its own total expected return $R_k = \sum_{t=0}^{T} \gamma^t r_{k,t}$ where $T$ is the time horizon.

## 2.2 Multi-Agent Deep Deterministic Policy Gradient

MADDPG (Lowe et al., 2017) is a multi-agent extension of the DDPG algorithm (Lillicrap et al., 2015). It adapts the CTDE paradigm, where each agent $i$ possesses its own deterministic policy $\mu^{(i)}$ for action selection and critic $Q^{(i)}$ for state-action value estimation, respectively parameterized by $\theta^{(i)}$ and $\phi^{(i)}$. All parametric models are trained off-policy from previous transitions $\zeta_t := (s_t, \mathbf{a}_t, \mathbf{r}_t, s_{t+1})$ uniformly sampled from a replay buffer $\mathcal{D}$. Each centralized critic is trained to estimate the expected return for a particular agent $i$ from the Q-learning loss:

$$\mathcal{L}^{(i)}(\phi^{(i)}) = \mathbb{E}_{\zeta_t \sim \mathcal{D}}\left[\left\|Q^{(i)}(s_t, \mathbf{a}_t; \phi^{(i)}) - \left[r_{i,t} + \gamma Q_{\text{target}}^{(i)}(s_{t+1}, \boldsymbol{\mu}(s_{t+1}))\right]\right\|^2\right] \tag{1}$$

Each policy is updated to maximize the expected discounted return of the corresponding agent $i$ :

$$J_{PG}^{(i)}(\theta^{(i)}) = \mathbb{E}_{\mathcal{D}}\left[Q^{(i)}(s_t, \mathbf{a}_t)\right] \tag{2}$$

Notice that while optimizing an agent's policy, all agents' observation-action pairs are taken into consideration. By that, the value functions of all agents are trained in a centralized, stationary environment, despite happening in a multi-agent setting. Moreover, this procedure allows for the learning of coordinated strategies, yet needs to be augmented with efficient exploration methods that reward novel action configurations which may lead to the discovery of higher-return behaviors.

## 3 Methods

### 3.1 Basic Influence

Intuitively, one can define coordination in a team of agents as the behavior of each individual agent being informed by other agents. Furthermore, agents' behaviors can be inter-affected either directly through communication for example or indirectly through task-specific shared goals and/or rewards or the dynamics of the environment. We hypothesize that when agents learn in a cooperative setting, they tend to affect each other's exploitation processes, we confirm the hypothesis throughout the paper and build on that to formalize a general method to foster influential interactions and learn meaningful coordination protocols. Specifically, consider a setting where $N$ agents with policies $\pi_1, \pi_2, \ldots, \pi_N$. We assume that the transition dynamics are dependent on agents' actions (i.e. $T(s'|s, \mathbf{a}) \neq T(s|s'))$, and use tasks that satisfy this assumption to validate our methods. We will write $\boldsymbol{\pi}, \boldsymbol{\pi}_{-j}$ to denote the joint policy of all agents and of all agents other than agent $j$, respectively. For two fixed agents $i$ and $j$, the influence of the behavior of $j$'s policy on $i$'s performance (expected return) can be quantified by "marginalizing" out all effects induced by other agents on the expected outcomes of $i$. More precisely, if the desired quantity is $q_i^{\pi_j}$, then for all $(s, a_j) \in S \times A_j$:

$$q_i^{\pi_j}(s, a_j) = \underset{\mathbf{a}_{-j} \sim \boldsymbol{\pi}_{-j}}{\mathbb{E}}\left[r_i(s, \mathbf{a})\right] + \gamma \sum_{s'} p(s'|s, a_j) \cdot \underset{a_j' \sim \pi_j(.|s_j')}{\mathbb{E}}\left[q_i^{\pi_j}(s', a_j')\right]$$
$$p(s'|s, a_j) = \sum_{\mathbf{a}_{-j}} T(s'|s, \mathbf{a}) \prod_{k \neq j} \pi_k(a_k|s) \tag{3}$$

One can note that, analogously to the action-value function $Q_i$, $q_i^{\pi_j}$ measures tuple values w.r.t $i$ where the learning dynamics of $i$ conditioned only on the behavior of $j$. Naturally, in expectation, $q_i^{\pi_j}$ represents the sum of rewards that an agent $i$ is expected to receive when influenced by $\pi_j$. Based on above, we say that an agent is an optimal influencer when agents only influenced by its policy learn optimal behaviors. To that end, if we fix an agent (say 1), then its influence, $F(\pi_1)$, on the rest of the team is defined as follows:

$$F(\pi_1) = \sum_{2 \leq i \leq N} \alpha_i \cdot \mathbb{E}_{s, a_1 \sim \pi_1}\left[\left(q_i^{\pi_1}(s, a_1) - \max_{\mathbf{u}} Q_i(s, \mathbf{u})\right)^2\right] \tag{4}$$

where $(\alpha_i)_{2 \leq i \leq N}$ are positive scaling factors that sum up to 1 and $Q_i$ is the action value function of agent $i$. Intuitively, since $F(\pi_1)$ is the difference between the expected rewards of other agents conditioned on the current behavior of $\pi_1$ and the maximum returns that can be achieved by other agents in the given task, minimizing $F$ will encourage $\pi_1$ to take actions that lead other agents'

conditioned returns $(q_i^{\pi_1})_{2 \leq i \leq N}$ closer to their maximum unrestricted returns $(\max_{\mathbf{u}} Q_i(s, \mathbf{u}))$, which matches our definition of an optimal influencer. Hence, the objective functions of agents $i \in [N]$ are: $J_i(\theta_i) = \mathbb{E}_{\boldsymbol{\pi}} \left[ \sum_{0 \leq t \leq T} \gamma^t r_{i,t} \right] - \lambda F(\pi_1) \mathbf{1}_{i=1}$, where $\theta_i$ are the parameters of $\pi_i$ and $\lambda$ is a hyperparameter called the influence importance temperature.

**Theorem 1.** *(Influence Gradient)*

$$\nabla_{\theta_1} F(\pi_1) = \sum_{2 \leq i \leq N} \alpha_i \mathbb{E}_{s, \pi_1} \big[ \nabla_{\theta_1} \log \pi_1(a_1|s) g(s, a_1)^2$$
$$+ 2g(s, a_1) \mathbb{E}_{s, \pi_1} [\nabla_{\theta_1} \log \pi_1(a_1|s) q_i^{\pi_1}(s, a_1)] \big] \tag{5}$$

*where* $g(T) = g(s, a_1) = q_i^{\pi_1}(s, a_1) - \max_{\mathbf{u}} Q_i(s, \mathbf{u})$.

It has been shown empirically that the existence of one influencer can greatly improve the overall team performance. Thus, we only consider one influencer throughout our practical sections and upon that, we call our method "Asymmetric Learning for Influencing a Team of Agents (ALITA)".

### 3.1.1 INFLUENCE WITH SINGLE ESTIMATOR

Practically, we quantify the influence $F(\pi_1)$ by initializing a network $Q^{cen} : S \times A \to \mathbb{R}^{N-1}$ with parameters $\phi^{cen}$; the $i$-th component of $Q^{cen}(.; \phi^{cen})$ estimates $q_i^{\pi_1}$ by minimizing the following loss at each timestep $t$:

$$\mathcal{L}(\phi^{cen}) = \mathbb{E}_{(\mathbf{x}, \mathbf{a}, r, \mathbf{x}') \sim \mathcal{D}_t} \big[ \| Q^{cen}(\mathbf{x}, \mathbf{a}; \phi^{cen}) - \mathbf{y} \|^2 \big] \tag{6}$$

where $\mathbf{y} \in \mathbb{R}^{N-1}$, $\mathbf{y}_i = r_i(\mathbf{x}, \mathbf{a}) + \gamma Q_{\text{target}}^{(i)}(\mathbf{x}', \boldsymbol{\pi}(\mathbf{x}'))$; $Q_{\text{target}}^{(i)}$ is the target critic of agent $\pi_i$, and $\mathcal{D}_t = \{T \mid T = (\mathbf{x}, \mathbf{a}, r, \mathbf{x}') \subset \mathcal{B} \text{ and } \mathbf{x}_1 = s_{1,t}, \mathbf{a}_1 = a_{1,t}\}$, where $\mathcal{B}$ is a buffer storing all agents' transitions. The reason for choosing such definition of $\mathcal{D}$ is that when fixing the current information of $\pi$ (i.e. at timestep $t$) while updating $Q^{cen}$, we practically marginalize out all agents' experience (other than 1 and $i$). Later in this section, we show that this trick yields a fairly good approximation of $q_i^{\pi_1}$. After training $Q^{cen}$, we compute the influence $F(\pi_1)$:

$$F(\pi_1) = \mathbb{E}_{(\mathbf{x}, ., r, \mathbf{x}') \sim \mathcal{B}} \big[ \| Q^{cen}(\mathbf{x}, \boldsymbol{\pi}(\mathbf{x})) - \mathbf{z} \|^2 \big] \tag{7}$$

where $\mathbf{z} = [\max_{\mathbf{u}} Q_2(\mathbf{x}', \mathbf{u}), \ldots, \max_{\mathbf{u}} Q_N(\mathbf{x}', \mathbf{u})]^T$. Note that the second term in the expectation (i.e. the target vector $\mathbf{z}$) is set to be undifferentiable with respect to $\pi_1$'s parameters and thus does not propagate through its network.

### 3.1.2 INFLUENCE WITH MULTIPLE INDIVIDUAL ESTIMATORS

As seen earlier, agent $\pi_1$ estimates the gap between each agent's value and its target value by employing a single network. Another desirable approach is to additively decompose the centralized estimator $Q^{cen}$, i.e. to use multiple estimators where each estimator, namely $Q_{\text{clone}}^{(i)}$, individually calculates a fairly good approximation of $q_i^{\pi_1}$. To reduce computational costs and arrive at better estimates, each estimator's network is initialized with the parameters of the corresponding critic network at each episode (i.e. $Q_{\text{clone}}^{(i)} \leftarrow Q_i$). The training is carried out similarly to that of the single estimator setting,

$$\mathcal{L}(\phi^{(i)}) = \mathbb{E}_{(\mathbf{x}, \mathbf{a}, r, \mathbf{x}') \sim \mathcal{D}_t} \big[ \| Q_{\text{clone}}^{(i)}(\mathbf{x}, \mathbf{a}; \phi^{(i)}) - \mathbf{y}_i \|^2 \big] \tag{8}$$

The influence function could be expressed as:

$$F(\pi_1) = \mathbb{E}_{(\mathbf{x}, ., ., \mathbf{x}') \sim \mathcal{B}} \frac{1}{N-1} \sum_{i=1}^{N-1} \left\| Q_{\text{clone}}^{(i)}(\mathbf{x}, \boldsymbol{\pi}(\mathbf{x})) - \mathbf{z}_i \right\|^2 \tag{9}$$

**Which approach yields better estimates of the true value of the influence?** The influence of an agent $\pi$ on a team of agents $T$ was defined as a measure of the improvement in the performance of $T$ given the current behavior of $\pi$. However, measuring the influence using function approximators might result in inaccurate estimates. To resolve this concern, we plot the influence estimates of the two prior approaches over time while they learn on the Cooperative Navigation MAPE task (Mordatch

& Abbeel, 2018), where the number of agents is $N = 6$. In Figure (1), we graph the average influence estimates over 40000 episodes and compare it to the true value. The latter is averaged over 1000 episodes following the current policies of agents and is reported every 5000 episodes. The plots show a relatively small bias of both methods during learning. However, as Figure (1) suggests, measuring influence using multiple individual estimators yielded more accurate values after enough training which substantiates its superiority over the shared network approach. Note that, although confirms our prior hypothesis, this experiment does not reflect the importance of employing the influence on the final performance of the agents as we will discuss that in Section (4).

### 3.2 Intrinsic Motivation for Diversified Team Behavior

In this section, we introduce a framework for achieving cooperative exploration by ensuring that agents are consistently tilted towards visiting under-explored state-action configurations; we start by providing a simple demonstration which shows that the number of environment steps required for all agents to randomly traverse all possible action configurations increases at least exponentially with the number of agents.

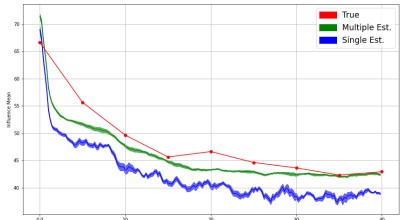

**Proposition 1.** *Consider an $L$-action setting of $n$ agents. In expectation, the number of steps $T$ needed to visit all $L^n$ action configurations at least once without coordinated exploration grows at least exponentially with the number of agents. More concretely, $\mathbb{E}[T] = \Omega(nL^n)$.*

Figure 1: Empirical evaluation of the bias in the proposed methods of measuring influence. The influence values estimated by single and multiple estimators are compared to the true values of $F$. The results for the estimated values are averaged across 8 runs.

To mitigate this issue, we assign agent $\pi_1$ a prediction error as an intrinsic reward to facilitate recognizing and learning novel behaviors:

$$r_t^{(\pi_1)} = r_{1,t}^{\text{ext}} + \lambda_\pi \underbrace{\|\phi(s_{1,t}, a_{1,t}) - (s_{1,t}, a_{1,t})\|^2}_{\psi(s_{1,t}, a_{1,t})} \qquad (10)$$

Where $\phi$ is an autoencoder network regularly trained on data generated by the policy $\pi_1$ and $\lambda_\pi$ is a hyper-parameter that balances the extrinsic and intrinsic reward terms. $\psi$'s expression stems from the observation that when an autoencoder is trained on data from a particular distribution, it will be good at reconstructing data from that distribution, while it will perform poorly if the data is from a different distribution. Thus, by employing $\psi$ as an intrinsic bonus, $\pi_1$ rewards states' observations and actions that do not belong to the data generated by it. In practice, $\phi$ is designed to be a relatively large network since we want it to be slightly overfitted to the training data so that it will not accidentally generalize to behaviors that we may deem novel (Fayad & Ibrahim, 2021; Zhang et al., 2019).

Nevertheless, assigning each agent a $\psi$ is not sufficient as it makes the case equivalent to independent exploration approaches. Thus, we propose a coordinated exploration method that takes into account other agents' behaviors, encouraging agents to diversify team behavior while maintaining good performance.

Specifically, we assign agents $\{\pi_i\}_{i=2}^N$ intrinsic penalty defined as:

$$r_{\pi_i}^{\text{int}}(s_t, \mathbf{a}_t) = -\exp\big(-\omega_i \psi(s_{1,t}, a_{1,t})\big) \|\pi_i(s_{i,t}) - a_{i,t}\|^2 \qquad (11)$$

This reward term aims at teaching the agents to recognize previous behaviors and synchronously select novel configurations. To build intuition, consider a case where $N = 2$. Whenever $(\pi, \mu)$ select an action tuple in the neighborhood of a frequently-visited tuple $(a_1, a_2)$ in a global state $s$, $\psi$ will be relatively small and the penalty, $r_\mu^{\text{int}}$, will be large. Conversely, if $(\pi, \mu)$ encounter a novel tuple, say $(a_1', a_2')$ in $s$, the small penalty of $\mu$ (Eq. (11)) together with a large reward for $\pi$ (Eq. (10)) will drive both agents to further explore this encounter.

In all, Fig. (2) shows how this framework can be augmented with the basic influence introduced earlier to reinforce learning and discovering coordinated behaviors.

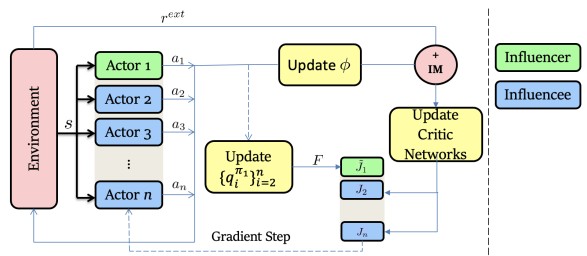

Figure 2: Architecture of the general proposed method

## 4 EMPIRICAL EVALUATION & ANALYSIS

The goals of our experiments are to: a) verify the performance of our method on a comprehensive set of multi-agent challenges (SMAC, MAPE, sparse-reward settings, and continuous control environments); b) perform ablations to examine which particular components of the proposed framework are important for good performance.

### 4.1 COOPERATIVE & MIXED GAMES

#### 4.1.1 STARCRAFT MULTI-AGENT CHALLENGE

StarCraft provides a rich set of heterogeneous units each with diverse actions, allowing for extremely complex cooperative behaviors among agents. We thus evaluate our method on several SC micromanagement tasks from the SMAC[1] benchmark (Samvelyan et al., 2019), where a group of mixed-typed units controlled by decentralized agents needs to cooperate to defeat another group of mixed-typed enemy units controlled by built-in heuristic rules with "difficult" setting; the battles can be both symmetric (same units in both groups) or asymmetric. Each agent observes its own status and, within its field of view, it also observes other units' statistics such as health, location, and unit type (partial observability); agents can only attack enemies within their shooting range. A shared reward is received on battle victory as well as damaging or killing enemy units. Each battle has step limits set by SMAC and may end early. We consider 5 battle maps grouped into **Easy** (2s3z), **Hard** (5m_vs_6m, 3s_vs_5z), and **Super Hard** (corridor, 3s5z_vs_3s6z) against 8 baseline methods using their open-source implementations based on PyMARL (Samvelyan et al., 2019): IQL (Tan, 1993), VDN (Sunehag et al., 2018), QMIX (Rashid et al., 2018), LIIR (Individual Intrinsic Rewards) (Du et al., 2019), LICA (Implicit Credit Assignment) (Zhou et al., 2020b), and MAVEN (Variational Exploration) (Mahajan et al., 2019), EDTI (Decision-Theoretic Exploration) , EITI (Information-Theoretic Exploration) (Wang et al., 2019).

The corridor map, in which 6 Zealots face 24 enemy Zerglings, requires agents to make effective use of the terrain features and block enemy attacks from different directions. A properly coordinated exploration scheme applied to this map would help the agents discover a suitable unit positioning quickly and improve performance, while 2s3z requires agents to learn "focus fire" and interception. For the asymmetric 5m_vs_6m, basic agent coordination alone such as "focus firing" no longer suffices (Du et al., 2019) and consistent success requires extended exploration to uncover complex cooperative strategies such as pulling back units with low health during combat. The 3s_vs_5z scenario features three allied Stalkers against five enemy Zealots. Since Zealots counter Stalkers, the only winning strategy for the allied units is to kite the enemy around the map and kill them one after another, causing the failure of independent learning algorithms to learn good policies in this task. The 3s5z_vs_3s6z is an extended scenario of 3s_vs_5z where 3 Stalkers and 5 Zealots battle against 3 Stalkers and 6 Zealots—The extra enemy makes this scenario far more challenging. For this task, we randomly chose 1 S and 1 Z to act as influencers because of the heterogeneity of the team. In the rest, agents are symmetric, hence the influencer is randomly chosen at the beginning of training.

For most of these scenarios, ALITA consistently shows the best performance with significant learning speed which confirms the effectiveness of our proposed methods. Detailed results are reported in Figure (3) as they present the median win rate during the training across 12 random runs.

---
[1] https://github.com/oxwhirl/smac

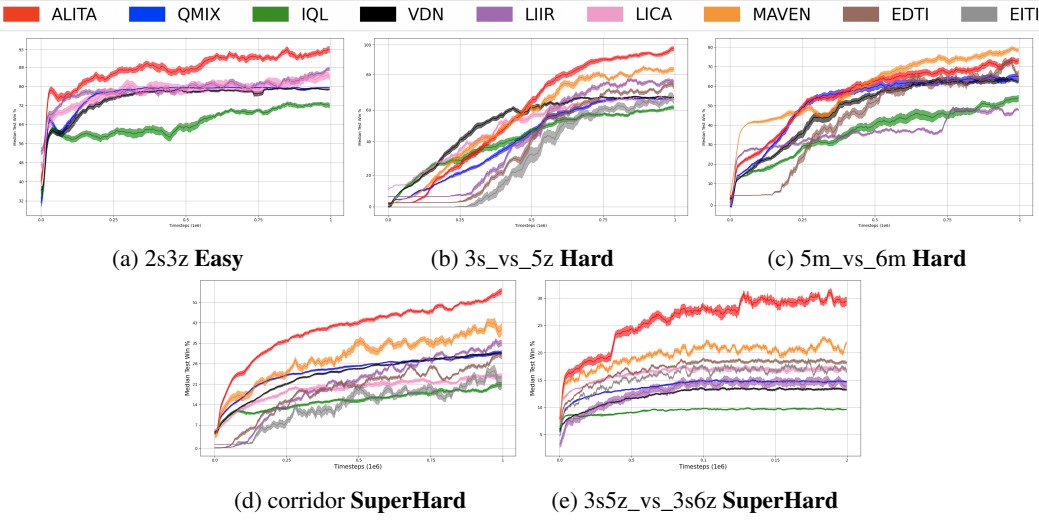

(a) 2s3z **Easy**  (b) 3s_vs_5z **Hard**  (c) 5m_vs_6m **Hard**

(d) corridor **SuperHard**  (e) 3s5z_vs_3s6z **SuperHard**

Figure 3: The median test win % of various methods across the SMAC scenarios.

### 4.1.2 SPARSE-REWARD SETTINGS

We test on two additional tasks to show the effectiveness our method on sparse-reward settings and compare it to famous influence-based coordinated exploration algorithms (Table (1)): EDTI, EITI (Wang et al., 2019), and Social Influence (Jaques et al., 2019).

**Sparse Push-Box:** A $15 \times 15$ room is populated with 2 agents and 1 box. Agents need to push the box to the wall in 300 environment steps to get a reward of 1000. Moreover, both can observe the coordinates of their teammate and the location of the box. However, the box is so heavy that only when two agents push it in the same direction at the same time can it be moved a grid. Agents need to coordinate their positions and actions for multiple steps to earn a reward.

**Sparse Secret Room:** A $25 \times 25$ grid is divided into three small rooms on the right and one large room on the left where 2 agents are initially spawned. There is one door between each small room and the large room. A switch in the large room controls all three doors. A switch also exists in each small room which only controls the room's door. The agents need to navigate to one of the three small rooms, i.e. the target room, to receive positive reward. The task is considered solved if both agents are in the target room. The state vector contains $(x, y)$ locations of all agents and binary variables to indicate if doors are open.

Table 1: Results on the **Push-Box** and **Secret Room** tasks after 20000 updates across 10 runs.

| Agents | Push-Box | | Secret Room | |
|---|---|---|---|---|
| | **Team Performance** | **Performance std** | **Avg Success Rate** | **Success Rate std** |
| ALITA | **146.66** | 34.13 | **0.68** | 0.04 |
| EDTI | **135.84** | 45.20 | 0.34 | 0.02 |
| Social Influence | 86.67 | 65.81 | 0.25 | 0.10 |
| EITI | 75.09 | 78.54 | 0.46 | 0.06 |

A notable reason for the good performance of ALITA on the two tasks is that through the intrinsic rewards, agents tend to explore the majority of the possible encounters at the beginning of learning which is crucial for estimating the value of the influence. Exploiting the latter, agents tend to reinforce learning jointly rewarding configurations.

### 4.1.3 MULTI-AGENT PARTICLE ENVIRONMENTS

To understand how the proposed method helps agents achieve cooperative behavior in nonstationary settings, we conduct experiments on the grounded communication environment [2] proposed in

---

[2] https://github.com/openai/multiagent-particle-envs

(Mordatch & Abbeel, 2018; Lowe et al., 2017). The chosen tasks are the **Cooperative Navigation, Cooperative Communication,** and **Physical Deception**.[3] We trained with 10 random seeds and reported results in Tables (2, 3, 4).

## 4.2 CONTINUOUS ENVIRONMENTS

To confirm the scalability of our algorithm to large continuous settings, we measure the performance of our algorithm on a suite of PyBullet (Tan et al., 2018) continuous control tasks, interfaced through OpenAI Gym (Brockman et al., 2016). Gym environments, however, are mainly single-agent settings, thus to evaluate our approach, we reframe the problem by introducing an additional learning agent that acts as an auxiliary agent. Crucially, both agents work collaboratively in order to find a region of the solution space where an agent accumulates higher rewards. We use TD3 (Fujimoto et al., 2018) as our learning model and test it against state-of-the-art algorithms in 5 gym environments. Our algorithm outperforms all baselines across all different environments (e.g. our method attains 131% return of SAC final performance on Humanoid-v3). For detailed results, see Appendix C.

## 4.3 ABLATIONS

We further investigate the significance of each component along with a symmetric extension of the proposed framework. Specifically, we consider the three cases: 1) **No F**: where the influence function does not contribute to the update rule to any of the policies; 2) **No IM**: where a randomly selected agent maximizes both the expected sum of $extrinsic$ rewards along with the influence function, and other agents' policies are learned using the DDPG; 3) **Symmetric**: where all agents simultaneously play the role of an influencer and influencee: they learn to maximize an augmented reward function ($extrinsic$ and $intrinsic$) along with the influence function.

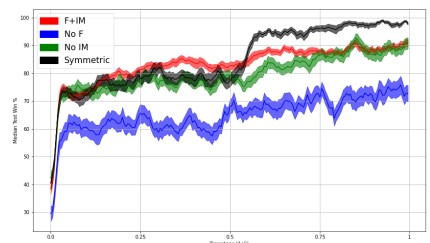

Figure 4: Ablations for different components of our framework on 2s3z scenario.

Results of the experiments conducted on the 2s3z SMAC scenario show that, in the absence of the intrinsic rewards (**No IM**), the agents experience a slightly decreased overall performance when compared to the significant decline induced by detaching the influence function (**No F**). In Figure (4), we observe that the agents following the **Symmetric** approach learn faster, and achieve a significantly higher median win rate. This approach, however, doubles the computational costs which restricts its applicability in larger settings.

## 5 RELATED WORK

We discuss recently developed methods for exploration in RL using intrinsic motivation, coordination in multi-agent RL, and influence-based coordinated exploration methods subsequently.

Intrinsic motivation (IM) has been increasingly used both in single-agent RL and multi-agent RL. A core idea of IM is to encourage the agent to take new actions or visit new states, thus exploring the environment and obtaining more diverse behaviors. One common approach is to approximate state or state-action visitation frequency and add a reward bonus to states the agent rarely covers (Tang et al., 2017; Bellemare et al., 2016; Martin et al., 2017). A more related IM approach is to evaluate state visitation novelty (Klissarov et al., 2019; Han et al., 2020; Burda et al., 2019) or state-action visitation novelty (Fayad & Ibrahim, 2021). Inspired by the latter, we provided a natural extension for this approach to the MARL settings by the learning of a "novelty" function. Other works make use of single-agent IM to construct their multi-agent intrinsic reward (Du et al., 2019; Iqbal & Sha, 2019). Each agent in (Du et al., 2019) learns a distinct intrinsic reward so that the agents are stimulated differently, even when the environment only feedbacks a team reward. This reward helps distinguish the contributions of the agents when the environment only returns a collective reward. In (Iqbal & Sha, 2019), each agent has a novelty function that assesses how novel an observation is to it, based

---

[3]Full description in Appendix B

on its past experience. Their multi-agent intrinsic reward is defined based on how novel all agents consider an agent's observation. A recent work (Liu et al., 2021) assigns agents a common goal while exploring. The goal is selected from multiple projected state spaces via a normalized entropy-based technique. Then, agents are trained to reach this goal in a coordinated manner.

Many works studied the cooperative settings in MARL; a straightforward approach is to use independent learning agents (fully decentralized learning). This approach, however, is shown to perform inadequately both with $Q$-learning (Matignon et al., 2012) and with policy gradient (Lowe et al., 2017). Therefore, we considered the CTDE paradigm, where each agent's policy takes its individual observation as many real life applications dictate, while the centralized critic permit for sharing of information during training. Policy gradient methods have been commonly used along with the CTDE paradigm in MARL, either by implementing a single centralized critic for all agents (Foerster et al., 2018), or one centralized critic for each agent (Lowe et al., 2017). Adopting the latter, we enable agents with different reward functions to learn in competitive and mixed scenarios as well.

Some other works encouraged cooperative interactions between agents by sharing useful information (Yang et al., 2020; Hostallero et al., 2020). In Hostallero et al. (2020), each agent broadcasts a signal that represents an assessment of the effect of the joint actions that all agents take on its expected reward. Different from our approach, this signal encourages agents to behave as is expected of them and does not benefit exploration. As for Yang et al. (2020), each agent learns an incentive function that rewards other agents based on their actions. Each agent's function aims to alter other agents' behavior to maximize its extrinsic rewards. To accomplish this, each agent requires access to every other agent's policy, incentive function, and return making this approach difficult to scale and execute. Additionally, Roy et al. (2019) proposed two policy regularizers approaches to promote coordination in a team of agent, one of which assumes that an agent must be able to predict the behavior of its teammates in order to coordinate with them, while the other supposes that coordinated agents collectively recognize different situations and synchronously switch to different sub-policies to react to them.

Similarly to our work, (Jaques et al., 2019) proposed a similar idea of rewarding an agent for having a casual influence on other agents' actions. Their method showed interesting results in terms of learning coordinated behavior. However, this casual influence is designed to reward policies for influencing other policies' actions without considering the "quality" of this influence. Barton et al. (2018) propose causal influence as a way to measure coordination between agents, specifically using Convergence Cross Mapping (CCM) to analyze the degree of dependence between two agents' policies. Our method also draws inspiration from the work of (Wang et al., 2019), as they define an influence-based intrinsic exploration bonus, called Value of Interaction (VoI), by the expected difference between the action-value function of one agent and its counterfactual action-value function without considering the state and action of the other agent. Particularly, the latter measures the expected behavior of an agent in a situation where it is not influenced by the other agent. Consequently, by maximizing the VoI, agents tend to explore meaningful interaction points as the distance between their action-value functions conditioned on other agents and their independent action-value (only conditioned on self behavior) functions is maximized. The primary difference from ALITA lies in our definition of the influence.

## 6 CONCLUSIONS & FUTURE WORK

We introduced a novel multi-agent RL algorithm for achieving coordination through assessing the influence an agent has on other agents' behaviors. Additionally, we proposed to learn an intrinsic reward for each agent to promote coordinated team exploration. We tested our algorithm on a wide variety of tasks with many challenges, such as partial observability, sparse rewards, and large spaces; these tasks include, but not limited to, SMAC, MAPE, as well as OpenAI gym continuous environments. Our methods achieved noticeable improvement over prominent algorithms on all tasks. One promising extension of our algorithm is to use Graph Attention Networks (Veličković et al., 2017; Zhou et al., 2020a) to learn the importance of the influencer in determining the influencees' policies and to establish a message-passing architecture in networked systems. The investigation of the effectiveness of these methods is left for future works.

## 7 REPRODUCIBILITY STATEMENT

We have provided an illustration of the proposed algorithm in Fig. (1) along with implementation details and hyperparameters selection in Appendix D. Furthermore, code is submitted with Supplementary Material and each algorithm is evaluated at least 10 times using random seeds on all environments.

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

## A  MATHEMATICAL DETAILS

**Theorem 1.** *(Influence Gradient)*

$$\nabla_{\theta_1} F(\pi_1) = \sum_{2 \leq i \leq N} \alpha_i \mathbb{E}_{s, \pi_1} \big[ \nabla_{\theta_1} \log \pi_1(a_1|s) g(s, a_1)^2 + 2g(s, a_1) \mathbb{E}_{s, \pi_1} [\nabla_{\theta_1} \log \pi_1(a_1|s) q_i^{\pi_1}(s, a_1)] \big]$$

where $g(T) = g(s, a_1) = q_i^{\pi_1}(s, a_1) - \max_{\mathbf{u}} Q_i(s, \mathbf{u})$.

*Proof.* We start by finding $\nabla_{\theta_1} q_i^{\pi_1}$:

$$\nabla_{\theta_1} q_i^{\pi_1}(s, a_1) = \gamma \mathop{\mathbb{E}}_{\substack{s' \sim p(.|s,a_1) \\ a_1' \sim \pi_1(.|s_1')}} \big[ \nabla_{\theta_1} \log \pi_1(a_1'|s') q_i^{\pi_1}(s', a_1') + \nabla_{\theta_1} q_i^{\pi_1}(s', a_1') \big]$$

Let $T = (s, a_1) \in S \times A_1$, and $\phi(T) = \gamma \mathbb{E}_{s', a_1'} [\nabla_{\theta_1} \log \pi_1(a_1'|s') q_i^{\pi_1}(s', a_1')]$ where the distribution of $s'$ is conditional on $s$ and $a_1$. Thus, write $\nabla_{\theta_1} q_i^{\pi_1}(T)$ as:

$$\phi(T) + \gamma \mathbb{E}_{s', a_1'} [\nabla_{\theta_1} q_i^{\pi_1}(T')] \qquad (T' = (s', a_1'))$$
$$= \phi(T) + \gamma \mathbb{E}_{s', a_1'} [\phi(T') + \gamma \mathbb{E}_{s'', a_1''} [\nabla_{\theta_1} q_i^{\pi_1}(T'')]] \qquad \text{(new tuples } T'' \text{are conditioned on previous ones)}$$
$$\vdots \qquad\qquad\qquad\qquad \text{(repeatedly unroll } \nabla q_i^{\pi_1})$$
$$\overset{*}{=} \sum_{T'} \sum_{k=0}^{\infty} \Pr(T \to T'; k) \gamma^k \phi(T')$$
$$\propto \sum_{T'} \underbrace{\lim_{t \to \infty} \Pr(S_t = s'|T, \gamma \pi_1)}_{d_T(s')} \phi(T') \qquad \text{(normalize the series term)}$$
$$= \sum_{T'} d_T(s') \mathbb{E}_{s'', a_1''} [\nabla_{\theta_1} \log \pi_1(a_1''|s'') q_i^{\pi_1}(s'', a_1'')] \qquad (d_T \text{ is the discounted visitation distribution induced by } \pi_1)$$
$$= \mathop{\mathbb{E}}_{\substack{s' \sim d_T(.) \\ a_1' \sim \pi_1(.|s')}} [\nabla_{\theta_1} \log \pi_1(a_1'|s') q_i^{\pi_1}(s', a_1')]$$
$$= \mathop{\mathbb{E}}_{T' \sim \rho(.|T)} [\nabla_{\theta_1} \log \pi_1(a_1'|s') q_i^{\pi_1}(T')] \qquad \text{(use } T' \text{ to simplify notation)}$$

Since $g(T) = q_i^{\pi_1}(s, a_1) - \max_{\mathbf{u}} Q_i(s, \mathbf{u})$, the gradient of $F(\pi_1)$ can be expressed as:

$$\nabla_{\theta_1} F(\pi_1) = \sum_{2 \leq i \leq N} \alpha_i \mathbb{E}_T \big[ \underbrace{\nabla_{\theta_1} \log \pi_1(a_1|s) g(T)^2}_{A} + 2g(T) \nabla_{\theta_1} q_i^{\pi_1}(T) \big]$$
$$= \sum_{2 \leq i \leq N} \alpha_i \mathbb{E}_T \big[ A + 2g(T) \mathop{\mathbb{E}}_{T' \sim \rho(.|T)} [\nabla_{\theta_1} \log \pi_1(a_1'|s') q_i^{\pi_1}(T')] \big]$$
$$= \sum_{2 \leq i \leq N} \alpha_i \mathbb{E}_T \big[ A + 2g(T) \cdot \mathbb{E}_{T'} [\nabla_{\theta_1} \log \pi_1(a_1'|s') q_i^{\pi_1}(T')] \big]$$

In all,

$$\nabla_{\theta_1} F(\pi_1) = \sum_{2 \leq i \leq N} \alpha_i \mathbb{E}_{s, \pi_1} \big[ \nabla_{\theta_1} \log \pi_1(a_1|s) g(s, a_1)^2 + 2g(s, a_1) \mathbb{E}_{s, \pi_1} [\nabla_{\theta_1} \log \pi_1(a_1|s) q_i^{\pi_1}(s, a_1)] \big]$$

$$\square$$

**Proposition 1.** Consider an $L$-action setting of $n$ agents. In expectation, the number of steps $T$ needed to visit all $L^n$ action configurations at least once without coordinated exploration grows at least exponentially with the number of agents. More concretely, $\mathbb{E}[T] = \Omega(nL^n)$.

*Proof.* Let $M = L^n$. Since agents tend to visit different action configurations with no coordinated behavior, one can equivalently say that agents uniformly pick a configuration out of all $L^n$ possible

configurations at each step. Let $T_k$ be the number of steps to visit the $k$-th distinct configuration after covering $k-1$ distinct action tuples. Observe that:

$$\mathbb{E}[T] = \sum_{k=1}^{M} \mathbb{E}[T_k] \tag{12}$$

Now, $\Pr[T_k = i] = \left(\frac{k-1}{M}\right)^{i-1}\left(1 - \frac{k-1}{M}\right)$ meaning that $T_k$ follows a geometric distribution. Thus,

$$\mathbb{E}[T_k] = \sum_{i=1}^{\infty}\left(\frac{k-1}{M}\right)^{i-1}\left(\frac{M-k+1}{M}\right)i = \frac{M}{M-k+1} \tag{13}$$

Getting back to Eq. (12),

$$\mathbb{E}[T] = M\sum_{k=1}^{M}\frac{1}{M-k+1} = M\sum_{k=1}^{M}\frac{1}{k} > M\int_{1}^{M}\frac{1}{x}dx = M\ln M = nL^n \ln L \tag{14}$$

And the conclusion follows. $\qquad\square$

## B  DETAILS OF MAPE TASKS

**Cooperative Navigation:** In this environment, $N$ agents must collaborate to reach a set of $N$ landmarks with known positions. Agents are rewarded based on how far any agent is from each landmark, meaning that the agents learn to spread with each agent covering one landmark. The agents, which occupy a significant physical space, are aware of their relative positions to each other and are further penalized when colliding with each other.

Table 2: Avg # of collisions per episode and avg agent distance from a landmark in the **cooperative navigation** task, after 25000 episodes. we achieved optimal performance in the $N = 3$ case, with near-optimal performance in the $N = 6$ case, as agents focused more on not colliding with each other (lowest collision average)

| Agent $\pi$ | $N = 3$ | | $N = 6$ | |
|---|---|---|---|---|
| | **Average dist.** | **# collisions** | **Average dist.** | **# collisions** |
| ALITA | **1.559** | **0.185** | 3.349 | **1.294** |
| MADDPG | 1.767 | 0.209 | **3.345** | 1.366 |
| DDPG | 1.858 | 0.375 | 3.350 | 1.585 |

**Cooperative Communication:** Here, a stationary speaker must guide a listener in an environment consisting of three landmarks of differing colors. At each episode, one landmark of a particular color is set as a goal for the listener to be reached, however, only the speaker can observe which landmark the listener must navigate to. Moreover, The speaker can produce a communication output at each time step which is observed by the listener. The latter must navigate the environment to reach the correct landmark. Agents are collectively rewarded at the end of an episode based on the listener's distance from the correct landmark.

**Physical Deception:** This environment consists of $N$ agents and $N$ landmarks, with one landmark as the target of all agents. The agents are rewarded based on the distance of the closest agent to the target landmark, making it sufficient for only one agent to reach it. An adversary agent also tries to reach the target landmark, while the agents are penalized as it gets closer to the target. The adversary, however, does not know which landmark is the target and must deduce it from the agents' behavior. For that reason, agents must cooperate to trick the adversary by learning to cover all the landmarks. This task shows that our algorithm is applicable not only to cooperative interactions but to mixed environments as well.

In the cooperative communication and physical deception tasks, ALITA obtained the highest success rate across all baselines.

Table 3: Percentage of episodes where the agent reached the target landmark and average distance from the target in the **cooperative communication** environment, after 25000 episodes.

| Agent | Target reach % | Average distance |
|---|---|---|
| ALITA | **90.3%** | **0.093** |
| MADDPG | 84.0% | 0.133 |
| DDPG | 32.0% | 0.456 |
| DQN | 24.8% | 0.754 |
| Actor-Critic | 17.2% | 2.071 |
| TRPO | 20.6% | 1.573 |
| REINFORCE | 13.6% | 3.333 |

Table 4: Results on the **physical deception** task, with $N = 2$ cooperative agents/landmarks. Success (*succ* %) for agents (AG) and adversaries (ADV) is if they are within a small distance from the target landmark.

| Agent $\pi$ | Adversary $\pi$ | AG succ % | ADV succ % | $\triangle$ succ % |
|---|---|---|---|---|
| ALITA | MADDPG | **95.2%** | 45.1% | 50.1% |
| MADDPG | MADDPG | 94.4% | 39.2% | 55.2% |
| MADDPG | DDPG | 92.2% | 16.4% | 75.8% |
| DDPG | MADDPG | 68.9% | 59.0% | 9.9% |
| DDPG | DDPG | 74.7% | 38.6% | 36.1% |

# C  ADDITIONAL EXPERIMENTS ON CONTINUOUS ENVIRONMENTS

Since the formulation of $F$ needs a shared buffer, SAC and TD3 stand as the best off-policy candidates to be incorporated with our framework, as they have shown great performances on many benchmarks. SAC, however, uses stochastic policies in general which makes it infeasible to combine with the formulation of $r^{\text{int}}$. Therefore, we use TD3 as our learning model to measure its performance on a suite of PyBullet (Tan et al., 2018) continuous control tasks, interfaced through OpenAI Gym (Brockman et al., 2016). While many previous works utilized the Mujoco (Todorov et al., 2012) physics engine to simulate the system dynamics of these tasks, we found it better to evaluate our method on benchmark problems powered by PyBullet simulator since it is widely reported that PyBullet problems are harder to solve (Tan et al., 2018) when compared to Mujoco. Also, Pybullet is license-free, unlike Mujoco that is only available to its license holders.

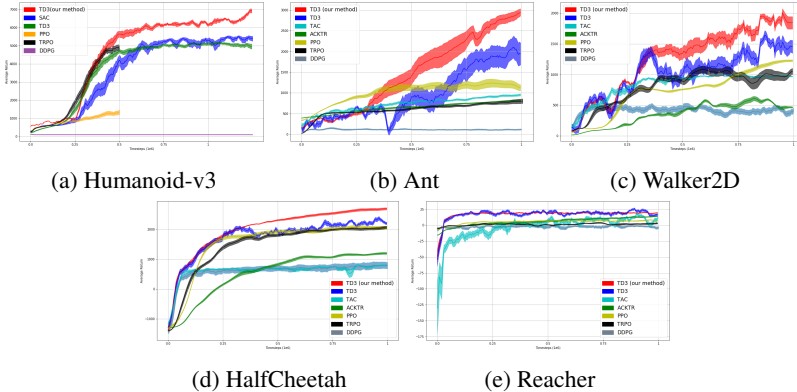

(a) Humanoid-v3     (b) Ant     (c) Walker2D

(d) HalfCheetah     (e) Reacher

Figure 5: Learning curves for the OpenAI gym continuous control tasks. The shaded region represents quarter a standard deviation of the average evaluation. Curves are smoothed for visual clarity.

We compare our method to the original twin delayed deep deterministic policy gradients (TD3) (Fujimoto et al., 2018); soft actor critic (SAC) (Haarnoja et al., 2018); proximal policy optimization (PPO) (Schulman et al., 2017), a stable and efficient on-policy policy gradient algorithm; deep deterministic policy gradient (DDPG); trust region policy optimization (TRPO) (Schulman et al.,

2015); Tsallis actor-critic (TAC) (Chen & Peng, 2019), a recent off-policy algorithm for learning maximum entropy policies, where we use the implementation of the authors[4][5]; and Actor-Critic using Kronecker-Factored Trust Region (ACKTR) (Wu et al., 2017), as implemented by OpenAI's baselines repository [6]. Each task is run for at least 1 million time steps and the average return of 15 episodes is reported every 5000 time steps. To enable reproducibility, each experiment is conducted on 10 random seeds of Gym simulator and network initialization. Results of the best performing agent of the two across different methods are reported in Figure (5).

# D  TRAINING DETAILS

## D.1  GENERAL CONFIGURATIONS

We use a buffer-size of $10^6$ entries and a batch-size of $1024$. We collect $100$ transitions by interacting with the environment for each learning update. For all tasks in our hyper-parameter searches, we train the agents for $15,000$ episodes of $100$ steps and then re-train the best configuration for each algorithm-environment pair for twice as long ($30,000$ episodes) to ensure full convergence for the final evaluation. We use a discount factor $\gamma$ of $0.95$, an influence importance temperature $\lambda$ of $0.1$, and a gradient clipping threshold of $0.5$ in all experiments unless otherwise specified. Each cloned critic is updated 4 time per step.

## D.2  SPARSE PUSH BOX AND SPARSE SECRET ROOM, MAPE, & GYM

We use the Adam optimizer (Kingma & Ba, 2014) to perform parameter updates. All models (actors, critics and proxy critics) are parametrized by feedforward networks containing two hidden layers of 128 units excpet for the autoencoder network where we use 7 hidden layers with dimensions (128, 64, 12, 3, 12, 64, 128), respectively. All models' parameters are initialized using Glorot Initialization method (Glorot & Bengio, 2010); while the autoencoder's parameters are initialized using Kaiming method (He et al., 2015). We employ the Rectified Linear Unit (ReLU) as activation function and layer normalization (Ba et al., 2016) on the pre-activations unit to stabilize the learning.

Table 5: Best found hyper-parameters for the Sparse-reward tasks, MAPE, & Gym environments

| HYPER-PARAMETER | PUSH BOX | SECRET ROOM | MAPE | HUMANOID-V3 | GYM (EXCEPT FOR HUMANOID-V3) |
|---|---|---|---|---|---|
| $\lambda_\pi$ | 0.10 | 0.10 | 0.01 | 0.10 | 0.10 |
| $\{\omega_i\}_1^{n-1}$ | 0.10 | 0.10 | 0.01 | 0.10 | 0.10 |
| $\beta$ | 0.15 | 0.10 | 0.10 | 0.15 | 0.1 |

## D.3  SMAC

The architecture of all agent networks is a DRQN (Hausknecht & Stone, 2015) with a recurrent layer comprised of a GRU with a $64$-dimensional hidden state, with a fully-connected layer before and after. All neural networks are trained using RMSprop ($\alpha = 0.99$ with no weight decay or momentum) with learning rate $5 \times 10^{-4}$.

Table 6: Best found hyper-parameters for the SMAC environments

| Hyper-parameter | Corridor | 5m_vs_6m | 3s_vs_5z | 2s3z |
|---|---|---|---|---|
| $\lambda_\pi$ | 0.09 | 0.03 | 0.01 | 0.01 |
| $\{\omega_i\}_1^{n-1}$ | 0.03 | 0.03 | 0.01 | 0.01 |
| $\beta$ | 0.15 | 0.15 | 0.10 | 0.10 |

---

[4]`https://github.com/haarnoja/sac`
[5]`https://github.com/yimingpeng/sac-master`
[6]`https://github.com/openai/baselines`

