# OpenReview forum: "Influence-Based Reinforcement Learning for Intrinsically-Motivated Agents"
_ICLR.cc/2022/Conference — ICLR 2022 Submitted_

### Official Review · Reviewer_sFW9 · 2021-10-29

**Correctness:** 2
**Technical Novelty And Significance:** 1
**Empirical Novelty And Significance:** 2
**Recommendation:** 3
**Confidence:** 4

**Main Review:**

This paper needs significant improvements in terms of better motivation and conceptual justification of the proposed methods, and with better choice of baselines in experiments. Technical writing also needs improvements to clarity and precision.

The proposed methods may be based on sensible intuition that the authors had in mind, but this is not successfully conveyed in the author's explication and justification of their method. One symptom of insufficient justification is that a reader still wonders why certain choices were made and why alternatives were not chosen. There are many examples of such in this paper:
1. It is not clear which agent, or how many agents in a homogeneous team, should be chosen to be the influencer labelled $\pi$ that is distinguished from all the other agents $\mu_1,\dotsc, \mu_{n-1}$.
2. In section 3.1.1, the function $Q^{\text{cen}}$ is trained to predict the TD target that other agents compute using off-policy samples from a replay buffer. The function is fitted to data from some specific subset $\mathcal{D}$, which was experienced during certain periods of training. Why, then, would it make sense to use this function as part of the regularization term later in equation (4), where the data in buffer $\mathcal{B}$ may be completely out of the training distribution for $Q^{\text{cen}}$?
3. At convergence of a centralized critic, the TD target is equal in expectation to the Q-value. So why should equation (3) aim to predict the TD target rather than just the Q-value itself?
4. Continuing from the previous point, why do we even need to use that special buffer $\mathcal{D}$ that is restricted to experiences where one agent's input is constrained to a certain observation-action pair, to train a whole separate $Q^{\text{cen}}$? If we have a well-trained centralized critic, then simply feeding it data that satisfies that constraint is enough to get the TD target (which asymptotically equals the critic's output).
5. It is unclear how the "counterfactual rollout" is involved in computing eq (4) (also unclear whether that's even intended). Displaying pseudocode in the main paper will help.
6. The given explanation for the use of $F_{\pi}$ in equation (4) as a regularizer is unintelligible. $Q^{\text{cen}}$ was trained to predict TD targets of other agents on a specific subset of experience. $F_{\pi}$ measures the difference between the predictions and the TD targets on some other set of experience. Minimizing $F_{\pi}$ (presumbly with respect to the parameters of $\pi$, but this is not stated explicitly) means to find $\pi$ whose behavior generates actions such that the pretrained $Q^{\text{cen}}$ has small prediction error compared to the actual TD targets of other agents. This affects $\pi$. This does not affect the other agents' policies $\mu_k$ in any direct way. It is unclear how changing $\pi$ to improve the prediction of $Q^{\text{cen}}$ is equivalent to changing $\pi$ to get other agents to "reach their goals faster and more efficiently".
7. Page 4, authors write "The influence of an agent $\pi$ on a team of agents T was defined as a measure of the improvement in the performance of T given the current behavior of $\pi$." One can have a large influence on a team and produce either signficant deprovement or improvement in performance. How, then, is "improvement in performance" a good measure of "influence"? Also, authors use the word "was", implying this definition was given previously. I can't find it.
8. "True value of influence" is mentioned on page 4 many times. This paper does not even provide a formal definition of "influence".
9. Again, the authors write "distance between the true q-value and the true q-target of each agent". This is zero in expectation.
10. The authors use the inability of a VAE to reconstruct data that is out of the training distribution, as a measure of the difference between seen states and new states. The authors should look at previous work on exploration bonuses that measure the novelty of states in far simpler and effective ways. Put simply, why should one use a VAE when one can use any distance metric?
11. equation (8) means that an agent with policy $\mu$ will still get a penalty (i.e., negative reward) for exploring new states. Why should $\pi$ be the one to receive positive intrinsic rewards rather than $\mu$?

This paper was written to address multi-agent exploration and influence. There are well-known multi-agent methods for exploration and influence [1,2,3], some even cited by this paper, that are not compared in the experiments. Authors say they compare to LIIR and LICA in the text, but don't show the results in the figures.

While the ablation experiments do show the impact of each component on overall performance, there are no results that allow one to conclude that a particular technical design choice has the specific effect for which it was designed. Examples: the regularizer in equation (4) is meant to generate more "influential behavior", but this concept is not defined and quantitatively measured.

Given all of these unclear design choices and lack of comparison to meaningful baselines, it doesn't even matter that the paper shows significant improvements versus standard MARL methods. No reader would be able to build on this work in its present form because of the disconnect between the technical method and the intuitive underpinnings.

More comments:
1. page 1, introduction, "dissimilarity between other agents' behaviors and their targets": It is unclear at this point of the paper what the authors mean by "targets". The most likely guess is some kind of "intended behavior". But actually it turns out on page 3 that the authors actually mean the temporal-difference target when they say "target". This was very confusing on the first read.
2. section 2.1, section header was used as a part of a sentence.
3. inconsistent notation. Agent is labeled first with index $k$, then later labeled by its policy $\pi$ or $\mu$.
4. In section 3.1, the authors write "Intuitively, one can define coordination in a team of agents as the behavior of each individual agent being informed by other agents". I can't tell whether the authors intended to give a definition of "coordination", or merely provide intuition .
5. Wrong figure labels on page 4, authors probably meant figure 1 rather than figure 2 when talking about results.



References
1. Mahajan, Anuj, et al. "MAVEN: multi-agent variational exploration." Proceedings of the 33rd International Conference on Neural Information Processing Systems. 2019.
2. Iqbal, Shariq, and Fei Sha. "Coordinated Exploration via Intrinsic Rewards for Multi-Agent Reinforcement Learning." arXiv preprint arXiv:1905.12127 (2019).
3. Jaques, Natasha, et al. "Social influence as intrinsic motivation for multi-agent deep reinforcement learning." International Conference on Machine Learning. PMLR, 2019.

**Summary Of The Paper:**

This paper is situated in the context of CTDE for cooperative MARL. The paper proposes new forms of intrinsic rewards to improve multi-agent exploration and a policy regularization term for an agent to have more influence over others. Experiments were conducted on existing benchmarks, such as StarCraft micromanagement and scenarios in the multi-agent particle environment, and two sparse-reward gridworld settings designed by the authors. The proposed method was compared with selected MARL baselines and ablations.

**Summary Of The Review:**

This paper needs significant improvements in terms of better motivation and conceptual justification of the proposed methods, and with better choice of baselines in experiments. Technical writing also needs improvements to clarity and precision.

---

> ### Author Response · Authors · 2021-11-23
> **Response to Reviewer sFW9**
>
> We thank the reviewer for their time and valuable comments. Please refer to the revised paper for details since we believe that most of your questions were answered in the main body.
>
> However, we address central misunderstandings here:
>
> 1. A primary concern is some design choices and lack of rigorous foundations of the method which raised multiple questions and concerns [comments 1-9], Please refer to section 3.1 as we believe all these issues have been resolved by introducing the definition and motivation of the influence. This brings us to the point where our method was confused with the TD-error: note that the definition of $q_i^{\pi_j}$ is quite similar to that of $Q$-value, which naturally means the update rules will be similar as well.
> 2. Comment 10: It is true that there are simpler exploration methods, however by checking the results in Table (1), you can see the comparison to some powerful exploration methods (EDTI and EITI) where both not only use specific architecture to improve exploration, but they also employ curiosity as an intrinsic reward term for all agents. In addition to that and upon adding comparisons to MAVEN, EDTI, and EITI on the SMAC scenarios, we believe that our exploration methods, although not fairly simple as claimed, are necessary for better performance. Also, please note that we do not use Variational AEs.
> 3. Comment 11: It does not matter whether $r_{\mu} ^ {\text{int}}$ is positive or negative as long as agent $\mu$ tends to prefer encounters that yield larger rewards. And since novel encounters mean greater $r^{\text{int}}_{\mu}$ values, $\mu$ is technically rewarded for exploring this encounter.
>
> We indeed added comparisons to LICA, LIIR, MAVEN (Social influence (Jaques et al 2019) is already included in case it has been missed), and an additional super hard scenario on SMAC. We hope that this resolves your concern about the choice of baselines.
>
> Overall, we believe we addressed most of your major concerns and we look forward to your response and questions.

---

> > ### Comment · Reviewer_sFW9 · 2021-11-29
> > **Significant revision to this paper deserves another independent review**
> >
> > I have seen the author's efforts during the rebuttal period to address the reviewer's questions. That effort has resulted in significant revision to the paper. Based on ICLR's review policy, I believe this new version of the paper deserves another round of independent review.

---

### Official Review · Reviewer_HYex · 2021-11-01

**Correctness:** 3
**Technical Novelty And Significance:** 3
**Empirical Novelty And Significance:** 2
**Recommendation:** 5
**Confidence:** 4

**Main Review:**

This problem domain is of considerable interest to a wide (and growing) audience. Hence, any substantial contribution has the potential for considerable impact. The paper demonstrates clear gains over baselines on a fairly wide array of popular benchmarks and considers a sizable set of useful baselines on at least a subset of these environments.

My primary critique of this paper is that the application of baselines is inconsistent across benchmarks in a way that makes it hard to say that a thorough head-to-head comparison has been done. In particular:

-Both the SMAC and sparse-reward benchmarks have useful benchmarks and are individually fairly thorough, but the set of baselines is disjoint between the two. Given the popularity of several of the approaches as baselines in the sparse reward setting, it would make sense to include them in the SMAC environments. My understanding is that their utility isn't limited to sparse reward settings -- if there is a good reason to exclude these (e.g. they have already been shown to be dominated by other baselines used in previous work, which I am not aware of), it would be important to say this.

-MAVEN (https://arxiv.org/abs/1910.07483) might be good to include.

-The Multi-Agent Particle Environment and Continuous Control experiments appear not to have any of these baselines and instead only have more standard CTDE and completely decentralized (assuming that to be the case, not certain) baselines.


My secondary critique is on the clarity of the work. Specifically:

-It's a bit hard to get the gestalt of the proposed method. My understanding is that one agent has an influencer regularizer to its optimization criterion, the influencer has reward (7), and the influencees have (8) added to their reward. It would be helpful if this were made more clear in  Figure 2, which appears not to differentiate how different agents receive intrinsic motivation, and has the term $\tilde J$, which I take to be the influence-regularized objective of the influencer, but I cannot see where this is explicitly defined. More minor, but I find (what at least I think is) the use of symbols for both policy networks and agent indexing confusing.

-An easy fix, but the method needs a name! Calling it DDPG(our method) in the results section is really confusing, especially in experiments that use vanilla DDPG. The figures have very small type.

-Might be just me, but I wish I had a bit better intuition for why precisely these definitions work well. It is not entirely clear to me as to why attempting to help other agents reach targets, precisely, is always useful...it would be helpful to build our intuition a little.


More minor, but in the contributions, it is stated that the intrinsic motivation amounts to an extension of RND. I get that it is similar to RND, but the choice made here, with an autoencoder, is distinct. Is it a better choice? It's possible to define a more RND-like CTDE extension, so I wouldn't call it precisely that.


**Summary Of The Paper:**

This paper introduces two techniques for Centralized Training with Decentralized Execution (CTDE) MARL. Specifically:
(1) It proposes an "influence" training objective by which one agent is encouraged to help the other agents reach target returns.
(2) It introduces intrinsic motivation and intrinsic cost terms aimed at encouraging efficient joint exploration of the Markov game.
It then goes on to give several empirical comparisons with baselines, showing the utility of the approach in Starcraft Multi-Agent Challenge (SMAC), two sparse reward settings, Multi-Agent Particle Environments, and OpenAI Gym continuous control environments. It also performs an ablation study in (SMAC), demonstrating the usefulness of each component of the proposed method.

**Summary Of The Review:**

In sum, I think the method has the potential for high impact, but it would be considerably strengthened with more thorough head-to-head comparison with popular methods -- that different baselines are limited to different benchmark suites considerably limits the results. I also have clarity concerns, but these are more minor and can easily be rectified with an edit.

---

> ### Author Response · Authors · 2021-11-23
> **Response to Reviewer HYex**
>
> We thank the reviewer for their time and valuable comments. Please see the revised paper for details.
>
> As suggested, we added the following baselines to the SMAC environments to solidify our experiments section: EDTI, EITI, MAVEN, LIIR, and LICA. One can note the clear gains in 4 out of 5 scenarios (2 of which are super-hard) over all baselines with very successful performance in the super-hard tasks.
>
> Regarding the clarity of the work, we included the rigorous mathematical formulation of our approach. Along with the revised notations, we believe that the updated version is more coherent, with a clear statement of our motivations.
>
> We also adhered to the suggested minor fixes (such as naming the method ALITA, and using clearer notations), with the learning objectives now stated explicitly in the main body.
>
> In all, we believe we addressed all your major concerns and we look forward to your response and questions.

---

> > ### Comment · Reviewer_HYex · 2021-11-23
> > **Thanks for the additional baselines!**
> >
> > Thank you for your response and revised draft. It is very useful to see the additional baselines added to the SMAC environment, which builds evidence to the utility of the method.
> >
> > I do think that this paper could benefit from more uniform deployment of baselines, still (though I realize that this is likely not feasible during the review period). In particular, MAVEN seems conspicuously absent from the other experiments, given how well it performs in SMAC. Social Influence would also be good to include, and broadly it still seems that the MAPE and continuous control experiments could really use these better baselines as well in order to really be convincing.
> >
> > Given separate concerns raised by other reviewers, I've decided not to change my score -- but I will pay attention to their responses and would consider editing based on that.

---

### Official Review · Reviewer_Pto8 · 2021-11-02

**Correctness:** 3
**Technical Novelty And Significance:** 2
**Empirical Novelty And Significance:** 2
**Recommendation:** 3
**Confidence:** 4

**Main Review:**

- Strengths:
	- The paper aims to develop a framework for achieving better exploration and coordination among agents in multi-agent settings, which is definitely an important and interesting research direction.
	- The proposed method is tested in a large number of multi-agent tasks.
- Weaknesses:
	- Many technical details of the proposed method are not very clear to me, which makes it very hard for me to judge if it is technically sound and how significant the contribution is (see my detailed comments below).
	- A lot of experimental results are not analysed at all in the paper. It is unclear to me how significant the experimental contributions are.
- Major concerns/questions:
	- Many notations used in the method section are not explained or quite confusing to me, which makes it very hard for me to understand the proposed method. For instance, what is $\mathbf{x}$ and $\mathbf{a}$ in Eq. (3)? The joint observation and action? What is the difference between $\mathbf{a}$ in Eq. (3) and $\boldsymbol{\mu}(\mathbf{x})$ in Eq. (4)? What is the difference between Eq. (3) and Eq. (4), if there is any? What is the difference between $\mathcal{D}$ and $\mathcal{B}$? One example of confusing notations is that $\pi$ was used to represent both an agent and a policy, so does $\mu_{i}$.
	- Many technical details of the proposed method are not explained/not clear to me. Here are just a few examples:
		- How is DDPG combined with the proposed method? The method section never mentions DDPG, but in the experimental section, there is suddenly DDPG (our method). A naive extension of DDPG to multi-agent settings would be independent DDPG, or do you use something similar to MADDPG? What is the loss function for DDPG (our method)? How do you train the actors and critics?
		- The authors mention that "$\phi$ is designed to be a relatively large network since we want it to be slightly overfitted to the training data so that it will not accidentally generalize to behaviors that we may deem novel." This is very vague. How do you ensure that the network is **slightly** overfitted? How do you ensure that it will not generalize to **novel** behaviours? How do you define novel here?
		- How do you combine the proposed influence function with the intrinsic reward in your method?  Figure 2 should be explained to make this clearer.
		- How should we select the influencer agent seems to be a critical issue for the proposed method, but this was never discussed in the paper (or maybe I miss it?). In fact, it's not even clear to me how do you select the influencer agent in the experiments.
	- For the SMAC results, the authors mention that they compare against baselines including LIIR and LICA, but I couldn't find them in the result (Figure 3). Also, the performance improvement on map corridor does not look as significant as it seems. If you look carefully into the y-axis, the proposed method achieves about 28% win rate, while the second best method achieves about 22% win rate.
   - For the results in Table 1-4, there seem to be no analysis/discussion at all. I think some discussion should at least be provided to explain why the proposed method can perform better than the baselines in some tasks.
   - In sparse Push-Box, for DDPG(our method), why is the average success rate 0.68 while the team performance is 146.66? I assume the team performance is the average total return. The reward function in this task seems to be that the agents get a reward of 1000 once succeed, otherwise no reward. Then, if the average success rate is 0.68, the average total return should be around 680. Can the authors clarify this?
	- Minor: the font size on the x and y axis in Figures 1, 3, and 4 is too small.

**Summary Of The Paper:**

This paper introduces the idea of estimating the influence an agent has on other agents' actions, in order to achieve better coordination among agents. An agent is chosen as the influencer, which estimates the gap between other agents' actions and their targets given its current action. The influencer is encouraged to minimize this gap to lead other agents closer to their target returns. The paper also proposes to learn an intrinsic reward for each agent to encourage agents to learn more diverse team behaviour. The proposed method was tested in a wide range of multi-agent tasks.

**Summary Of The Review:**

I vote for rejecting this paper as many technical details of the proposed method are not very clear to me and many experimental results in the paper are not analysed.

---

> ### Author Response · Authors · 2021-11-23
> **Response to Reviewer Pto8**
>
> We thank the reviewer for their time and valuable comments. Please see the revised paper for details.
>
> Regarding your concerns:
>
> 1. We understand the confusion our previous notations made, thus we used a unified and commonly-used notation to resolve this issue. We also included rigorous mathematical descriptions of the method which will hopefully make it more technically sound and easier to understand.
> 2. As for the learning model, we update the policies using MADDPG (without using ensembles of policies). Also, the learning objectives are now stated explicitly in the main body.
> 3. Regarding your questions about the autoencoder architecture and training, we ensure that it is slightly overfitted to the data by choosing an appropriate architecture and hyperparameters. These details are included in the appendix. Now, what is meant by “ensure that it will not generalize to novel behaviors”? As mentioned, we slightly overfit the autoencoder to the data so that if, during training, the policy encounters a novel data point, it will be able to identify that this point does not belong to the familiar policy distribution since the autoencoder is only good at reconstructing the data that it has been trained on (and slightly overfitted to). In all, bad reconstruction means novel data points [1]. Naturally, a novel data point is a state-action pair that was not encountered before or, equivalently, does not belong to the policy distribution.
> 4. In our environments, the symmetry of the agents in a team guarantees the validity of randomly choosing one agent to be the influencer throughout training and execution. Our main contribution, however, remains in the formulation of $F(\pi_1)$, and we regard the problem of dynamically choosing influencers as a fruitful area for future work.
> 5. Regarding your comment on the Push-Box environment, Table 1 contains the results of two different environments: Push-Box, and Secret Room. (The metric of the first is the average reward, while the second is the average success rate).
>
> In all, we believe we addressed all your major concerns and we look forward to your response and questions.
>
> [1] Yunbo Zhang, Wenhao Yu, and Greg Turk. Learning novel policies for tasks. In International
> Conference on Machine Learning, pp. 7483–7492. PMLR, 2019.

---

### Official Review · Reviewer_dZZd · 2021-11-02

**Correctness:** 2
**Technical Novelty And Significance:** 2
**Empirical Novelty And Significance:** 3
**Recommendation:** 5
**Confidence:** 4

**Main Review:**

STRENGTHS

The presented algorithm divides a group of agents into one “influencer” and other “influencees”, which is new and interesting. The implementation of influence-based regularization and curiosity-driven incentives is easy to implement (e.g., the influence function can use the one-step TD target directly), which is scalable for large-scale multi-agent systems. The empirical evaluation is extensive, in which the proposed algorithm achieves outperformance on a large set of challenging tasks, e.g., StarCraft II benchmark, MPE environments, and several gridworld tasks. This topic is of interest to the community because efficient exploration in MARL is a long-standing problem.

WEAKNESSES

The main weaknesses and concerns are listed as follows:

1) The core component of this paper, the formulation of influence functions (Eq. (4) and (6)), is not motivated. From the perspective of optimization in centralized training with decentralized execution (CTDE) paradigm, this regularization is equivalent to the one-step TD loss of the whole multi-agent system. Minimizing this influence-based regularization can be regarded as minimizing the empirical Bellman error during the centralized training process. Thus, it is not an additional component of the original cooperative MARL algorithms. The clearer motivation of this influence function is critical for this paper.


2) The connection to EITI/EDTI (Wang et al., 2019) needs to be justified more clearly. For example, EITI/EDTI introduces an influence-based multi-agent exploration, in which they formulate the influence of other agents by the concept of Value of Interaction (Vol), and they also utilize decentralized exploration method, i.e., count-based exploration. As EITI/EDTI is closely related to this paper, this paper needs to discuss it in more detail.

3) In general, the influencers of a multi-agent system may be dynamic during policy training or in different timesteps of an episode. This paper chooses the influencers without considering these factors. The author may justify in principle that this choice of influencers during policy learning is reasonable.

4) This paper does not have a thorough comparison to some SOTA MARL baselines, e.g., MAVEN (Mahajan et al., 2019), QPLEX (Wang et al., 2021), MAAC (Iqbal and Sha, 2019), DOP (Wang et al., 2021), and FOP (Zhang et al., 2021), in which MAVEN is a popular exploration method, QPLEX is a strong value-based baseline, MAAC, DOP and FOP are advanced actor-critic algorithms. The empirical part of this paper would become stronger by including these baselines.

Minor Question:

This paper claims that it extends random network distillation (RND) to the multi-agent setting for crafting a "novelty" metric, as shown in Eq. (7). In the RND, the curiosity-driven incentive on the timestep t is dependent on the prediction error of the next state and next action. However, in Eq. (7), the intrinsic reward on the timestep t is dependent on the current state and action. Why is the way to formulate curiosity-driven intrinsic rewards different from RND?

[1] Mahajan, Anuj, et al. "MAVEN: multi-agent variational exploration." Proceedings of the 33rd International Conference on Neural Information Processing Systems. 2019.

[2] Wang, J., Ren, Z., Liu, T., Yang, Y., and Zhang, C. QPLEX: Duplex dueling multi-agent q-learning. International Conference on Learning Representations, 2021.

[3] Iqbal, Shariq, and Fei Sha. "Actor-attention-critic for multi-agent reinforcement learning." In International Conference on Machine Learning, pp. 2961-2970. PMLR, 2019.

[4] Wang, Yihan, et al. "Off-policy multi-agent decomposed policy gradients." International Conference on Learning Representations, 2021.

[5] Zhang, Tianhao, et al. "FOP: Factorizing Optimal Joint Policy of Maximum-Entropy Multi-Agent Reinforcement Learning." International Conference on Machine Learning. PMLR, 2021.


**Summary Of The Paper:**

This paper introduces a new exploration method for cooperative multi-agent reinforcement learning (MARL), which utilizes influence-based regularization and curiosity-driven incentives to encourage coordinated and diverse exploration. This paper formulates the dissimilarity between other agents' behaviors and their one-step TD targets as the influence metric and extends the random network distillation (RND) to the multi-agent setting for crafting a "novelty" metric. Empirical results show that this method achieves improved performances on a comprehensive set of challenging tasks.

**Summary Of The Review:**

This paper proposes a new method for MARL exploration and demonstrates extensive empirical performance on a large set of tasks. However, the reviewer has major concerns on the core component of this paper, i.e., the motivation of influence functions. In addition, the reviewer thinks that the proposed method has some implicit assumptions and the discussion and comparison of related work need to be improved.

---

> ### Author Response · Authors · 2021-11-23
> **Response to Reviewer dZZd**
>
> We thank the reviewer for their time and valuable comments. Please see the revised paper for details.
>
> Regarding your concerns:
>
> 1. The confusion between the TD-error and our influence function was addressed by the concrete formulating of the influence function with an explicit presentation of our assumptions.
> 2. The links between our paper and EITI/EDTI (Wang et al., 2019) are precisely discussed in the related work section.
> 3. In our environments, the symmetry of the agents in a team guarantees the validity of randomly choosing one agent to be the influencer throughout training and execution. Our main contribution, however, remains in the formulation of $F(\pi_1)$, and we regard the problem of dynamically choosing influencers as a fruitful area for future work.
> 4. Given the limited time, we managed to include three additional baselines: MAVEN, LICA, LIIR, tested on SMAC environments. In addition, we ran the preexisting spare-reward-task baselines (EDTI, EITI) on SMAC (# of overall baselines = 9).
>
> Overall, we believe we addressed all your major concerns and we look forward to your response and questions.

---

> > ### Comment · Reviewer_dZZd · 2021-11-30
> > **Response to the rebuttal**
> >
> > Thank the authors for the detailed response and additional experiments, which clarified some of my concerns. However, my major concern on the core component of this paper, i.e., the motivation of influence functions, still remains. Therefore, I will raise my score to 5.

---

### Author Response · Authors · 2021-11-23
**Comment To All Reviewers**

Dear Reviewers,

Our former presentation of the paper seems to have caused multiple misunderstandings about our original methodology which raised many concerns. We have updated the paper accordingly to address all the rigor, clarity, significance, and intuition issues.

The main changes in the updated version of our paper:
1.  A clear definition of the problem and influence function, giving mathematical foundation to ALITA (our method's name) + presenting the update rule with respect to the regularized objectives.
2.  Unified notation across the whole paper to avoid any misunderstandings.
3.  A wider set of experiments (+1 superhard SMAC task) on the existing baselines, together with three additional baselines: MAVEN, LICA, LIIR, as suggested, in addition to further analysis of the empirical results.
4. A deeper discussion of the connections between our work and the work of (Wang et al., 2020).

Tonghan Wang, Jianhao Wang, Yi Wu, and Chongjie Zhang. Influence-based multi-agent exploration. ICLR 2020

---

### Decision · Program_Chairs · 2022-01-20

**Decision:**

Reject

**Comment:**

At this time, this work is not yet ready for publication. The core idea--influence functions--was poorly explained in the initial submission, and although major changes to the paper were made to rectify this, at least some reviewers of the remain unconvinced and it is unclear that the paper has been fully evaluated with this confusion resolved. There are a sufficient number of other concerns around the paper, that having rectified these more fully and outside the tight time constraints of the rebuttal period, I hope for an interesting resubmission in future.